# RaanA: A Fast, Flexible, and Data-Efficient Post-Training Quantization Algorithm

## Abstract

Post-training Quantization (PTQ) has become a widely used technique for improving inference efficiency of large language models (LLMs). However, existing PTQ methods generally suffer from crucial limitations such as heavy calibration data requirements and inflexible choice of target number of bits. In this paper, we propose RaanA, a unified PTQ framework that overcomes these challenges by introducing two novel components: 1) RaBitQ-H, a variant of a randomized vector quantization method RaBitQ, designed for fast, accurate, and highly efficient quantization; and 2) AllocateBits, an algorithm that optimally allocates bit-widths across layers based on their quantization sensitivity. RaanA achieves competitive performance with state-of-the-art quantization methods while being extremely fast, requiring minimal calibration data, and enabling flexible bit allocation. Extensive experiments demonstrate RaanA's efficacy in balancing efficiency and accuracy.

## 1 Introduction

The rapid advancement in deep learning has demonstrated that increasing the size of neural networks (NNs) often leads to significant improvements in performance across various tasks (Kaplan et al., 2020; Betker et al., 2023; Achiam et al., 2023; Grattafiori et al., 2024). Large language models (LLMs), such as GPT (Brown et al., 2020; Achiam et al., 2023) and LLaMA (Touvron et al., 2023; Grattafiori et al., 2024), have exhibited remarkable capabilities, but their deployment remains a challenge due to substantial computational and memory requirements. Among the various strategies developed to improve LLM inference, Post-Training Quantization (PTQ) has emerged as a widely adopted approach for optimizing the memory usage and inference speed of LLMs, balancing between accuracy and efficiency (Frantar & Alistarh, 2022; Frantar et al., 2023; Lin et al., 2024; Guan et al., 2024; Lee et al., 2024; Chee et al., 2023). The essential idea of PTQ is to reduce the numerical precision of the parameters of a trained model, replacing high precision floating-point representations with lower precision alternatives, which in turn reduces memory consumption and alleviates memory bandwidth constraints.

Optimal Brain Quantizer (OBQ) is one of the earlier influential methods in PTQ. It formulates the quantization problem as an optimization task, where the core objective is to minimize the layer-wise quantization error $\widehat{\boldsymbol{W}}^{(\ell)*} = \arg\min_{\widehat{\boldsymbol{W}} \in \mathcal{C}} \left\| \boldsymbol{X}^{(\ell)} \boldsymbol{W}^{(\ell)} - \boldsymbol{X}^{(\ell)} \widehat{\boldsymbol{W}} \right\|_{\mathcal{F}}^2$, where $\boldsymbol{X}^{(\ell)}$, $\boldsymbol{W}^{(\ell)}$ and $\widehat{\boldsymbol{W}}$ are the input feature, original weight matrix and quantized weight matrix of layer $\ell$ (a linear transformation layer) respectively. This approach has led to significant improvements in quantization accuracy by optimizing the quantization parameters (rescale, zero-point etc.) according to layer-wise behavior approximation. OBQ has become a foundation of PTQ research, with many recent methods building upon its framework (Frantar et al., 2023; Lee et al., 2024; Lin et al., 2024; Chee et al., 2023). Despite their effectiveness, existing approaches based on the OBQ framework typically suffer from notable limitations:

- First, the OBQ framework only controls the error of the output of each layer and does not account for the overall error in the model output; more importantly, it treats all layers identically, whereas in practice, their importance can vary: the error in the first layer propagates through all subsequent layers, while that in the last layer affects only itself. Omitting this hierarchy in error sensitivity can lead to suboptimal quantization strategies.

- Second, the optimal solution of the OBQ framework highly depends on the input $\boldsymbol{X}^{(\ell)}$ which, however, is not constant across batches, and as a result, all the quantization methods based on this framework require a lot of calibration data to accurately estimate the input features (more precisely, the so-called layer-wise Hessian $\boldsymbol{X}^{(\ell)\top}\boldsymbol{X}^{(\ell)}$) (Hassibi et al., 1993). This high requirement of data not only limits the efficiency of quantization, but also the generalization performance of the method, as it can perform worse if the data distribution is far from its calibration data distribution.

- Moreover, the OBQ framework directly replaces each weight matrix with a quantized weight matrix. Since we are only concerned with the output of a layer rather than its internal structure, there is potential for greater flexibility by allowing additional operations within a layer.

The above-mentioned limitations are not exclusive to OBQ-based methods, but also shared by many other PTQ methods. For example, Norm-Tweaking (Li et al., 2024), although only a plugin for adjusting LayerNorm parameters, also highly relies on calibration data, and EasyQuant (Tang et al., 2024), although it does not require calibration, only controls layer-wise error and is limited to entry-wise float-point rounding. In this paper, we propose a novel PTQ framework that overcomes the above issue, based on the following ideas.

**RaBitQ-H: Adopting Efficient Vector Quantization Methods to LLM Quantization**    If we don't stick to preserving the structure of the original matrix multiplication and embrace a broader class of operations, it is possible to bring in much stronger tools from vector quantization. Specifically, in this paper we adopt RaBitQ (Gao & Long, 2024; Gao et al., 2024), a state-of-the-art vector quantization algorithm that enjoys very fast computation and well-controlled error-bound. However, RaBitQ is originally designed to solve approximate nearest neighbor search (ANNS) and can not be directly adopted to LLM quantization (see the detailed explanation in Section 5). To address this issue, in this paper we propose RaBitQ-H, a variant of RaBitQ, to address the challenges of adopting RaBitQ to LLM quantization.

**AllocateBits: Allocating Bits Unevenly Across Layers**    As mentioned before, it is suboptimal to uniformly assign computational resources across all layers, as some layers can be more important than others. There has been a research direction called Mixed Precision Quantization (MPQ), which allows different numbers of bits used in each layer (Pandey et al., 2023; Guan et al., 2024; Behtash et al., 2025). However, existing MPQ methods are mostly based on heuristics and are typically limited to binary choices of bit-widths (e.g. 4-bits v.s. 8-bits), making them not sufficiently flexible. In this paper, we propose a simple and effective approach to estimate the importance of each layer and formulate the bit allocation task as an integer programming problem that allows arbitrary bit-width choices. By solving this integer program, we obtain an optimal bit allocation strategy across layers.

In this paper, we propose **RaanA**: a novel PTQ framework combining **RaBitQ-H and AllocateBits**. RaanA enjoys the following superiority:

1. Extremely Fast and Device-Independent: Unlike other state-of-the-art PTQ methods that generally require hours or even days to perform on large models, RaanA generally only requires tens of minutes on 70b models. Additionally, RaanA is largely device-independent, with most of its computation can be efficiently performed on CPUs, reducing its reliance on GPU devices.

2. Minimal Calibration Required: RaanA requires only a few or even zero samples for calibration, as opposed to most existing methods that typically require a huge amount of data.

3. Strong Performance: RaanA performs comparably to state-of-the-art PTQ methods. Notably, it remains effective even at extremely low bit-widths (e.g., $< 3$ average bits), a regime where many other methods struggle.

4. High Flexibility: RaanA allows any choice of target average number of bits, enabling more flexible and and fine-grained quantization choices.

**Paper Structure**    In Section 2 we introduce some background on the methods we use and related work. Sections 3 to 5 describe RaanA in detail: Section 3.1 presents the preliminaries and overall

framework; Section 4 explains how to estimate the sensitivity of each layer and solve the bit allocation problem; and Section 5 introduces the RaBitQ-H algorithm. In Section 6, we present our experimental results. Finally, in Section 7, we conclude the paper and discuss potential future directions.

## 2 BACKGROUND AND RELATED WORK

In this section, we introduce some of the related works as well as the background of methods we use in this paper.

**Weight Quantization in Post-training Quantization**   Among many methods that aims to make large language models more efficient, PTQ targets at improving the efficiency in inference-time. In this paper we especially focus on weight quantization. The problem of weight quantization in PTQ can be described as follows: given a pre-trained large language model, find another model who requires much less bits to store such that it approximates the behavior of the original model. OBQ (Frantar & Alistarh, 2022) is one of the early works for PTQ. As mentioned in Section 1, many existing PTQ methods follows the framework of OBQ and is devoted to better or faster solve the layer-wise OBQ problem (Frantar et al., 2023; Lee et al., 2024; Lin et al., 2024; Shao et al., 2023; Chee et al., 2023; Tseng et al., 2024). As we mentioned in Section 1, most of these methods highly relies on performing float-point rounding and estimating the layer-wise Hessian.

**Vector Quantization for Approximate Nearest Neighbor Search**   There have been a vast literature of vector quantization in ANNS. The classical scalar quantization (SQ) and its variant LVQ (Aguerre-bere et al., 2023) take finite uniform grids as codebooks and independently round every coordinate to an unsigned integer. These methods support to compute inner product based on arithmetics of unsigned integers without decompression. However, they failed to achieve reasonable accuracy using low-bit quantization ($< 4$ bits). PQ (Jegou et al., 2011) and OPQ (Ge et al., 2013) construct a codebook via KMeans clustering and quantizes vectors by mapping them to the nearest vector in the codebook. With clustering, these methods better capture the distributions of a dataset and provide better accuracy using low-bit quantization. However, their computation of inner product replies on looking up tables, which is costly in modern systems. RaBitQ (Gao & Long, 2024) and its multi-bit extension (Gao et al., 2024) construct a codebook by translating, normalizing, and randomly rotating finite uniform grids. They achieve superior accuracy while enabling efficient computation. Theoretically, they provide an unbiased estimator with asymptotically optimal error bounds for the space-accuracy trade-off in inner product estimation between unit vectors. See Section A.2 for more details.

**(Randomized) Hadamard Transformation**   In this paper, one critical technique we used is Ran-domized Hadamard Transformation (RHT), which is a randomized version of Hadamard Trans-formation and can be viewed as an approximation of random rotation (Tropp, 2011). Hadamard Transformation, in our context, is a specific class of orthonormal matrix whose matrix multiplication that can be fast computed, and is used in many quantization works (Tseng et al., 2024; Ashkboos et al., 2024; Liu et al., 2024; Agarwal et al., 2024) and other areas. Due to space limitation, we defer a more detailed introduction of RHT to Section A.1.

## 3 PRELIMINARIES

Throughout this paper, we use bold lowercase letters to represent vectors (e.g. $\boldsymbol{x}$) and bold uppercase letters to represent matrices (e.g. $\boldsymbol{A}$), and we use the corresponding unbold letters with subscript to represent the entries of vectors or matrices (e.g. $x_i$ is the $i$-th entry of vector $\boldsymbol{x}$).

In this paper, we fix a trained neural network $f : \mathbb{R}^{n \times d_0} \times \mathbb{R}^m \to \mathbb{R}$ as the object of study, where $d_0$ is the dimensionality of the input and $m$ is the number of learnable parameters[1]. We use $\boldsymbol{x} \in \mathbb{R}^{n \times d_0}$ and $\boldsymbol{\theta}$ to denote the input data and the collection of all parameters of $f$, respectively, where $n$ is the token number[2].

A NN model typically consists of multiple linear transformation blocks where the input is multiplied by a weight matrix. These include, for example, feed-forward layers as well as the Q, K, and V

---

[1]A neural network with multi-dimensional output can be viewed as several neural networks with scalar output, and therefore we only consider NNs with scalar output.

[2]$n$ can also be understood as the batch size, since we only consider linear layers that do not involve any cross-token interaction.

transformation layers in Transformer architectures (Vaswani et al., 2017). We assume the model contains a total of $L$ linear layers and denote the input features, parameter matrix, and output features of the $k$-th linear layer by $\boldsymbol{X}^{(k)} \in \mathbb{R}^{n \times d_k}$, $\boldsymbol{W}^{(k)} \in \mathbb{R}^{d_k \times c_k}$, and $\boldsymbol{H}^{(k)} = \boldsymbol{X}^{(k)} \boldsymbol{W}^{(k)} \in \mathbb{R}^{n \times c_k}$, respectively, where $d_k$ and $c_k$ are the input and output dimensions of the $k$-th linear layer. Note that $\boldsymbol{X}^{(k)}$ is principally a function of $\boldsymbol{x}$, and $\boldsymbol{W}^{(k)}$ is a function of $\boldsymbol{\theta}$, but we omit the arguments $\boldsymbol{x}$ and $\boldsymbol{\theta}$ when they are clear from context.

We use $\boldsymbol{x}_i^{(k)} \in \mathbb{R}^d$ to represent the $i$-th row of $\boldsymbol{X}^{(k)}$ and $\boldsymbol{w}^{(k)}i \in \mathbb{R}^d$ to represent the $i$-th column of $\boldsymbol{W}^{(k)}$. The $(i,j)$-th entry of $\boldsymbol{H}^{(k)}$ is given by $h^{(k)}i, j = \left\langle \boldsymbol{x}_i^{(k)}, \boldsymbol{w}_j^{(k)} \right\rangle$.

### 3.1 Overall Framework

We first present the overall framework of RaanA in Algorithm 1. It takes a trained model, calibration data and desired quantization parameters as input and output quantized weight matrix together with other information used for de-quantization (see Algorithm 3 for the de-quantization algorithm). The algorithm consists of two parts: determining each layer's target bit-width, and performing quantization. In Algorithm 1 we use the error estimator $\mathsf{Err}_k$ and vector quantization algorithm RaBitQ-H as black boxes, and they will be specified in the subsequent sections.

---

**Algorithm 1:** Overall Framework of RaanA

**Input:** trained model parameters $\boldsymbol{\theta}_0$, bit-width candidate set $\mathscr{B}$, overall bits budget $R$ and calibration data $\{\boldsymbol{x}_i\}_{i=1}^{n_c}$;

/* AllocateBits                                                         */

Solve the bits-allocation problem

$$
\{b_k^*\}_{k=1}^L = \arg \min_{\{b_k\}_{k=1}^L} \frac{1}{n_c} \sum_{k=1}^L \sum_{i=1}^{n_c} \mathsf{Err}_k (b_k; \boldsymbol{x}_i) \tag{1}
$$

$$
\text{s.t.} \quad \sum_{k=1}^L b_k m_k \leq R \text{ and } b_k \in \mathscr{B}, \forall k \in [L];
$$

/* Quantization                                                      */

Calculate $\left( \widehat{\boldsymbol{W}}^{(k)}, \boldsymbol{r}^{(k)}, \boldsymbol{D}^{(k)} \right) = \mathsf{RaBitQ\text{-}H} \left( \boldsymbol{W}^{(k)}, b_k^* \right)$ for $k = 1, 2, \cdots L$;

**Return:** $\left\{ \left( \widehat{\boldsymbol{W}}^{(k)}, \boldsymbol{r}^{(k)}, \boldsymbol{D}^{(k)} \right) \right\}_{k=1}^L$.

---

**Algorithm 1: The overall framework of RaanA.** $\mathsf{Err}_k(b; \boldsymbol{x})$ is an estimation of the overall error brought by the $k$-th linear layer with $b$-bit quantization, estimated at point $\boldsymbol{x}$, and RaBitQ-H is the RaBitQ-H quantization algorithm. $m_k = d_k c_k$ is the number of parameters at layer $k$; $n_c$ is the number of calibration data; $\mathscr{B}$ is a set containing all desired choice of layer-wise bit-width (e.g. $\mathscr{B} = \{1, 2, \cdots 8\}$), and $R$ is the desired total number of bits used (i.e. bits per parameter times total number of weight parameters).

## 4 AllocateBits: Error Estimation and Bits Allocation

In this section, we discuss how to calculate $\mathsf{Err}_k(b, \boldsymbol{x})$ and how to solve the bits-allocation problem eq. (1). We begin by defining this quantity. In the model $f$, if we replace the parameters of the $k$-th layer, $\boldsymbol{W}^{(k)}$, with a $b$-bit quantized version, this leads to an error in the output $\boldsymbol{H}^{(k)}$. We denote this error by $\boldsymbol{\epsilon}^{(k)}(b) \in \mathbb{R}^{n \times c}$. Consequently, $\boldsymbol{\epsilon}^{(k)}(b)$ leads to an error in the overall model output $f$. We define $\mathsf{Err}_k(b, \boldsymbol{x})$ as the absolute difference between the model outputs before and after quantization at layer $k$ with $b$ bits.

We first state a property of $\boldsymbol{\epsilon}^{(k)}(b)$ under RaBitQ-H, using Theorem 4.1, which is justified by the analysis of RaBitQ (see Section A.2 for more details).

**Assumption 4.1.** There exists a constant $K > 0$, such that for any $k \in [L], i \in [n], j \in [c_k]$, $\boldsymbol{\epsilon}^{(k)}(b) \in \mathbb{R}^{d_k}$ is a random vector such that

$$\boldsymbol{\epsilon}_{i,j}^{(k)}(b) \lesssim 2^{-b} \left\| \boldsymbol{x}_i^{(k)} \right\| \left\| \boldsymbol{w}_j^{(k)} \right\| \tag{2}$$

with high probability, where we use $\lesssim$ to hide constant terms.

When Theorem 4.1 holds, Theorem 4.2 is a direct corollary of Theorem B.2, which we prove in the appendix.

**Corollary 4.2** (Informal). *Fix the model input $\boldsymbol{x}$ and parameter $\boldsymbol{\theta}$, and suppose $\boldsymbol{\epsilon} = \boldsymbol{\epsilon}^{(k)}(b)$ satisfies Theorem 4.1, then the following statement holds with probability at least $0.99$:*

$$\mathsf{Err}_k\left(b, \boldsymbol{x}\right) \lesssim 2^{-b} \sqrt{\frac{\log c_k}{d_k}} \left\| \left. \frac{\partial f(\boldsymbol{\theta}, \boldsymbol{x})}{\partial \boldsymbol{H}^{(k)}} \right|_{\substack{\boldsymbol{x}=\boldsymbol{x} \\ \boldsymbol{\theta}=\boldsymbol{\theta}}} \right\|_{\mathcal{F}} \left\| \boldsymbol{X}^{(k)} \right\|_{\mathcal{F}} \left\| \boldsymbol{W}^{(k)} \right\|_{\mathcal{F}}, \tag{3}$$

*where we use $\lesssim$ to hide constant coefficients and small $O(1/d)$ terms.*

Denote $\alpha_k = \sqrt{\frac{\log c_k}{d_k}} \left\| \left. \frac{\partial f(\boldsymbol{\theta}, \boldsymbol{x})}{\partial \boldsymbol{H}^{(k)}} \right|_{\substack{\boldsymbol{x}=\boldsymbol{x} \\ \boldsymbol{\theta}=\boldsymbol{\theta}}} \right\|_{\mathcal{F}} \left\| \boldsymbol{X}^{(k)} \right\|_{\mathcal{F}} \left\| \boldsymbol{W}^{(k)} \right\|_{\mathcal{F}}$ be the coefficient in Theorem 4.2, we estimate the error introduced by $b$-bit quantization at the $k$-th layer by $\alpha_k 2^{-b_k}$. Then the bits-allocation problem eq. (1) can be rewritten as

$$\{b_k^*\}_{k=1}^L = \arg \min_{\{b_k\}_{k=1}^L} \sum_{k=1}^L \alpha_k 2^{-b_k}$$

$$\text{s.t. } \sum_{k=1}^L b_k m_k \leq R, \tag{4}$$

$$b_k \in \mathscr{B}, \forall k \in [L].$$

### 4.1 Solving the Bits-Allocation Problem

Now we consider solving problem eq. (4). At first glance, this is a nonlinear integer programming problem which is known to be NP-complete (Karp, 2010). Nevertheless, we note that the problem size is manageable in practice so that it can be efficiently solved via dynamic programming. Let $g = \mathsf{gcd}\left(m_1, \cdots m_L, R\right)$ be the greatest common divisor (GCD) of all $m_k$'s and $R$, then we have

$$\sum_{k=1}^L b_k m_k \leq R \iff \sum_{k=1}^L b_k \frac{m_k}{g} \leq \frac{R}{g}, \tag{5}$$

and thus the total bits budget can be reduced to $R/g$. Thanks to the design choice of most large language models, whose hidden sizes are usually powers of 2, in practice $g$ is typically very large. As a result, the size of this problem can be prominently reduced to a scale where finding the global optimum via a dynamic programming algorithm becomes feasible. We provide the algorithm for solving this problem in Section C.1.

Algorithm 4 from Section C.1 runs in $O\left(L|\mathscr{B}|R/g\right)$ time. In practice, both $L$ and $|\mathscr{B}|$ are less than 100, $\frac{R}{g}$ is on the order of $10^5$ and the entire process can be completed in a few seconds on CPU. For LLaMA models, the value of $g$ is on the order of $10^6$, making the divide-by-GCD trick – although seemingly simple – crucial for efficiency; without it, the algorithm would be millions of times slower.

### 4.2 Few-shot and Zero-shot Calibration

In Algorithm 1, a calibration set $\{\boldsymbol{x}_i\}_{i=1}^{n_c}$ is used to estimate the error (i.e. $\alpha_k$ in eq. (4)). In principle, we can certainly use a large amount of data to obtain a better estimator of $\alpha_k$. However, in RaanA, $\alpha_k$ only depends on the norm of input and the Jacobian of the overall output w.r.t. the output of layer $k$. Unlike the calibration in OBQ-based methods that requires estimating the layer-wise Hessian, these values here are practically very stable and can be estimated using a very small amount of data. This has also been observed in some previous work – for example, Khromov & Singh (2023) has found that the empirical mean of Jacobian quickly converges to the Lipschitz constant of the model.

In our implementation, we consider two settings: **few-shot calibration** and **zero-shot calibration**. In few-shot setting, we use 5 samples from the training set of the corresponding dataset, which is significantly less than what mainstream methods use (e.g. Quip# uses more than 6000 samples (Tseng et al., 2024)). In the zero-shot calibration setting, we only use one synthetic sentence to estimate $\alpha_k$, without resorting to any actual training data. Specifically, we repeat the following sentence 100 times to form our single calibration data point in the zero-shot setting [3]:

> "The curious fox leaped over the quiet stream, its reflection rippling in the golden afternoon light."

## 5   RABITQ-H: A VARIANT OF RABITQ WITH RANDOMIZED HADAMARD TRANSFORMATION

In Gao et al. (2024), the authors introduced RaBitQ, a universal multi-bit vector quantization algorithm that preserves the results of vector inner products and achieves an asymptotically optimal error rate. In large language models, the basic operation and bottleneck is matrix multiplication (MM), which can also be viewed as performing multiple inner products. Therefore, it is possible to adopt RaBitQ as our quantization method. RaBitQ has the appealing property that the error rate is controlled in all cases with high probability and does not require handling outliers (see details in Section A.2). This ensures the desired property stated in Theorem 4.1.

However, directly applying RaBitQ in the MM scenario can be inefficient. In RaBitQ, it requires applying a random rotation to each vector during the pre-processing stage. For a $d$-dimensional matrix, applying a random rotation takes $O(d^2)$ time. This is acceptable in the original ANNS scenario, where the data dimension is much smaller than the number of data points, making the rotation cost negligible. In contrast, in the context of large language models and MM, the number and dimensionality of the vectors are comparable[4], making the cost of performing or storing a random rotation comparable to that of the original MM, which negates the potential benefits of quantization. To address this issue, we replace the random rotation in the original paper with a Randomized

---

**Algorithm 2:** Preprocess: Quantization

**Input:** weight matrix $\boldsymbol{W}^{(k)} \in \mathbb{R}^{d_k \times c_k}$, desired number of bits $B \in \mathscr{B}$;
$\boldsymbol{\xi} \leftarrow \{\xi_i\}_{i=1}^{d_k}$ where $\xi_i$ is sampled i.i.d. from the Rademacher distribution;

$\boldsymbol{D}^{(k)} \leftarrow \text{diag}(\boldsymbol{\xi})$;

$\boldsymbol{W}' \leftarrow \text{Hadamard}\left(\boldsymbol{D}^{(k)}\boldsymbol{W}\right)$;

$\widehat{\boldsymbol{W}}^{(k)}, \boldsymbol{r}^{(k)} \leftarrow \text{RaBitQ}\left(\boldsymbol{W}', B\right)$;

**Return:** $\widehat{\boldsymbol{W}}^{(k)}, \boldsymbol{r}^{(k)}, \boldsymbol{D}^{(k)}$.

---

**Algorithm 3:** Inference: Matrix Multiplication Estimation

**Input:** input features $\boldsymbol{X}^{(k)} \in \mathbb{R}^{n \times d_k}$, quantized weight matrix $\widehat{\boldsymbol{W}}$, rescale factor $\boldsymbol{r}^{(k)} \in \mathbb{R}^{c_k}$, Rademacher samples $\boldsymbol{D}^{(k)} \in \mathbb{R}^{d_k}$, desired number of bits $B \in \mathscr{B}$;

$\boldsymbol{X} \leftarrow \text{Hadamard}\left(\boldsymbol{D}^{(k)}\boldsymbol{X}^{(k)\top}\right)^{\top}$;

$\boldsymbol{z} \leftarrow \frac{2^B-1}{2}\boldsymbol{X}\boldsymbol{1}$;

$\boldsymbol{Y} \leftarrow \boldsymbol{X}\widehat{\boldsymbol{W}}^{(k)}\boldsymbol{1}\boldsymbol{r}^{\top} - \boldsymbol{z}\boldsymbol{r}^{\top}$;

**Return:** $\boldsymbol{Y}$.

---

Algorithms 2 and 3: **Quantization and Inference algorithms of RaBitQ-H.** Here Hadamard refers to Hadamard transformation (see Section A.1 for a formal definition) and RaBitQ refers to the original RaBitQ algorithm *without random rotation*. Principally they are both vector operations, and here by applying them to matrices we mean applying column-wise. $\boldsymbol{r}^{(k)} \in \mathbb{R}^{c_k}$ in the algorithms is the rescale factor output by Extended RaBitQ, and $\boldsymbol{1}$ means a $c_k$-dimensional all-one vector.

Hadamard Transformation (RHT), which is known to approximate random rotation in many cases (Tropp, 2011). RHT has the following desired properties: 1) For each layer $k$, it only requires $d_k$ random bits, making storing the transformation not a bottleneck; 2) It can be efficiently computed in time $O((c_k + n)\log d_k)$ using existing Hadamard kernels (Agarwal et al., 2024). We adopt the analysis from the original RaBitQ paper (Gao et al., 2024) and confirm that the output of RaBitQ-H

---

[3]This sentence was suggested by ChatGPT.

[4]$c_k$ is the number of vectors and $d_k$ is the vector dimension.

| Method | Avg. bits | llama-7b | llama-13b | llama2-7b | llama2-13b | llama2-70b |
|--------|-----------|----------|-----------|-----------|------------|------------|
| fp16 | 16 | 5.68 | 5.09 | 5.47 | 4.88 | 3.31 |
| GPTQ | 2+ | 44.01 | 15.60 | 36.77 | 28.14 | - |
| AWQ | 2+ | 2.6e5 | 2.8e5 | 2.2e5 | 1.2e5 | - |
| OmniQuant | 2+ | 9.72 | 7.93 | 11.06 | 8.26 | 6.55 |
| Quip#* | 2 | 9.95 | 7.18 | 12.3 | **7.60** | 4.87 |
| RaanA | 2.1 | 13.70 | 8.28 | 18.31 | 51.05 | 4.81 |
| RaanA | 2.3 | **8.53** | **6.63** | **10.63** | 8.96 | **4.49** |
| GPTQ | 3+ | 8.81 | 5.66 | 6.43 | 5.48 | 3.85 |
| AWQ | 3+ | 6.35 | 5.52 | 6.24 | 5.32 | - |
| OmniQuant | 3+ | 6.15 | 5.44 | 6.03 | 5.28 | 3.78 |
| Quip#* | 3 | 6.29 | 5.52 | 6.19 | 5.34 | 3.71 |
| RaanA | 3.1 | 6.33 | 5.53 | 6.20 | 5.48 | 3.66 |
| RaanA | 3.3 | **6.10** | **5.38** | **6.00** | **5.27** | **3.59** |
| EasyQuant | 4+ | 6.01 | 5.29 | - | - | - |
| GPTQ | 4+ | 6.22 | 5.23 | 5.69 | 4.98 | 3.42 |
| AWQ | 4+ | 5.78 | 5.19 | 5.60 | 4.97 | - |
| OmniQuant | 4+ | **5.77** | **5.17** | **5.58** | **4.95** | **3.40** |
| Quip#* | 4 | 5.83 | 5.20 | 5.66 | 5.00 | 3.42 |
| RaanA | 4.1 | 5.86 | 5.20 | 5.69 | 5.02 | 3.42 |
| RaanA | 4.3 | 5.83 | **5.17** | 5.65 | 4.98 | **3.40** |

Table 1: **Perplexity results on wikitext2**. Methods labeled '+' in the number of bits use tricks such as grouping and keeping outliers full-precision that bring $0.1 \sim 0.3$ extra bit cost. Quip#* refers to Quip#$_{\text{no FT \& no } E_8}$. For OmniQuant, GPTQ and AWQ, we compare with the version with grouping size 128. Best performance of each category is labeled bold.

satisfies Theorem 4.1. The complete quantization and de-quantization algorithms are provided in Algorithms 2 and 3.

**Remarks on Implementation.** The practical implementation of RaBitQ-H is slightly more complicated than described here, involving two additional components. First, since the fast Hadamard transformation only applies to vector sizes that are powers of 2, we need to handle vectors whose sizes are not powers of 2. Second, we use small implementation tricks (such as input centralization) that preprocess the inputs and weights before running RaBitQ-H; these tricks do not affect the output of the quantization / de-quantization algorithm but in practice reduce the error rate. Due to space limitations, we defer these details to Sections C.2 and C.3.

# 6 EXPERIMENT RESULTS

In this section, we present our experimental results. We focus on the performance loss after quantization, compared to the full-precision (fp16) model. Following previous work (Lin et al., 2024; Shao et al., 2023), we evaluate our method on the LLaMA model family (Touvron et al., 2023) and the Qwen3 (Yang et al., 2025) model with language modeling tasks. Due to limited space, we only show results on LLaMA models in the main paper and defer results with Qwen3 to the appendix.

**Datasets.** As in previous studies (Lin et al., 2024; Shao et al., 2023), we evaluate our method on the wikitext2 (Merity et al., 2016) and c4 (Raffel et al., 2020) datasets. We split the test / validation sets into sequences of length 2048 as test samples and measure the average perplexity of the quantized model on each test sample. For c4, since the full validation set is too large and model performance converges quickly, we use 500 samples as the test set.

**Baseline Methods.** We compare RaanA with GPTQ (Frantar et al., 2023), AWQ (Lin et al., 2024), OmniQuant (Shao et al., 2023), and Quip#$_{\text{no FT \& no } E_8}$ (Tseng et al., 2024)[5]. Additionally, for 4-bit

---

[5]The full version of Quip# (Tseng et al., 2024) fine-tunes the model after quantization. In this paper, we compare RaanA with Quip#$_{\text{no FT \& no } E_8}$, the version without fine-tuning, since fine-tuning is a universal plug-in component in quantization and is orthogonal to our contribution.

quantization, we also compare with EasyQuant (Tang et al., 2024), a lightweight quantization method that does not require calibration data and targets 4-bit quantization. Note that most baseline models use tricks such as grouping and keeping full-precision outliers that introduce extra bit costs[6], generally ranging from 0.1 to 0.3 bits. To make a fair comparison, we report RaanA performance with $x + 0.1$ bits and $x + 0.3$ bits for $x \in 2, 3, 4$.

## 6.1 PERFORMANCE

In this sub-section, we compare the performance of RaanA under few-shot calibration with baseline methods. The results on wikitext2 are reported in Table 1 and we defer the results on c4 to Section D due to limited space. It is clear from the table that RaanA is comparable with or better than baseline methods, especially in the extreme regime of 2+ bits quantization.

## 6.2 ZERO-SHOT CALIBRATION VS FEW SHOT CALIBRATION

In this subsection, we compare the performance between few-shot calibration and zero-shot calibration. Results for wikitext2 are displayed in Table 2 for results on c4 are deferred to Section D. We also add results from EasyQuant, which also does not require calibration data, as a comparison. It is clear from the resutls that, although the performance does generally go down a little bit with zero-shot calibration, they are generally comparable with few-shot calibration results, validating the effectiveness of zero-shot calibration for RaanA.

| Method | Avg. bits | llama-7b | llama-13b | llama2-7b | llama2-13b | llama2-70b |
|---|---|---|---|---|---|---|
| fp16 | 16 | 5.68 | 5.09 | 5.47 | 4.88 | 3.31 |
| RaanA-few | 2.1 | 13.70 | 8.28 | 18.31 | 51.05 | 4.81 |
| RaanA-zero | 2.1 | 15.50 | 10.12 | 26.13 | 13.37 | 7.89 |
| RaanA-few | 3.1 | 6.33 | 5.53 | 6.20 | 5.48 | 3.66 |
| RaanA-zero | 3.1 | 6.45 | 5.63 | 6.41 | 5.55 | 4.01 |
| EasyQuant | 4+ | 6.01 | 5.29 | - | - | - |
| RaanA-few | 4.1 | 5.86 | 5.20 | 5.69 | 5.02 | 3.42 |
| RaanA-zero | 4.1 | 5.86 | 5.23 | 5.73 | 5.04 | 3.50 |

Table 2: **Perplexity Comparison Between Zero-shot Calibration and Few-shot Calibration on wikitext2**. RaanA-zero refers to RaanA with zero-shot calibration and RaanA-few refers to RaanA with few-shot calibration.

## 6.3 QUANTIZATION TIME

We note that due to the lack of GPU implementation of RaBitQ, the main part of RaanA is run on CPU, which is the time bottleneck. In our current implementation, the only parts requiring GPUs are calibration (which requires one or a few backward passes of the model) and Hadamard Transformation. Despite the main part running on CPU, RaanA still runs much faster than many existing quantization methods, demonstrating its high efficiency and device independence.

| Model | Time (s) |
|---|---|
| llama2-7b | 301.74 |
| llama2-13b | 567.61 |
| llama2-70b | 3293.26 |

Table 3: **Quantization Time**. The time required to complete the RaanA quantization process with few-shot calibration and average number of bits of 2.1.

We report the time RaanA used for quantization in Table 3 under a specific setting as an illustration of the efficiency of RaanA. The experiments are conducted with 4 NVIDIA A100 GPUs[7]. As shown, RaanA completes the quantization of a 70B model in under one hour, significantly faster than other heavyweight quantization methods such as Quip#$_{\text{no FT \& no } E_8}$, which can take up to 10 hours for the same model size, despite having comparable performance.

---

[6]The only exception in our baselines is Quip#$_{\text{no FT \& no } E_8}$, which has a precise control over the average number of bits. We admittedly require slightly more bits to match its performance. However, RaanA is significantly more lightweight than Quip#$_{\text{no FT \& no } E_8}$, which uses over 6000 calibration samples to estimate the layer-wise Hessian.

[7]Here the GPU configuration does not significantly impact the quantization time since the bottleneck is the CPU computation of RaBitQ. The machine we use has two AMD EPYC 7513 CPUs.

## 6.4 ABLATION STUDIES

In this subsection, we investigate the contribution of each component of RaanA by conducting experiments and analyses with individual components isolated. The results demonstrate that both RabitQ-H and AllocateBits play indispensable roles in the overall performance of RaanA.

**Ablation with AllocateBits** To examine the effect of AllocateBits, we compare the performance of the full RaanA model with that of a variant using uniform bit allocation (i.e., assigning the same number of bits to each parameter). The results, presented in Table 5, clearly show that AllocateBits significantly enhances performance—particularly in low-bit quantization—even when calibrated using only a single, arbitrarily chosen sentence.

|  | RabitQ | RabitQ-H | w/o quant |
|---|---|---|---|
| Extra Time | $\Theta(nd^2)$ | $\Theta(nd \log d)$ | $\Theta(ndc)$ |
| Extra bpm | $16d/c$ | $1/c$ | $16$ |

Table 4: **De-quantization time and extra bpm requirement comparison between RabitQ and RabitQ-H.** The "w/o quant" shows the original time and bpm required if without quantization, assuming the original model is using 16-bits float point number.

| Bits Allocation | Calibration | 2 bits | 3 bits | 4 bits |
|---|---|---|---|---|
| uniform | - | 1253.78 | 7.63 | 5.8 |
| AllocateBits | few-shot | 71.52 | 6.43 | 5.72 |
| AllocateBits | zero-shot | 46.33 | 6.60 | 5.77 |

Table 5: **Perplexity results under different bits allocation algorithms**. Experiments are performed with llama2-7b on wikitext2. Notice that since the calibration is only needed in AllocateBits stage, no calibration is needed after it is replaced with uniform bits allocation.

**Ablation with RabitQ-H** The RHT module in RabitQ-H primarily improves de-quantization efficiency by reducing both computation time and the number of additional bits required to store de-quantization information. To highlight this improvement, Table 4 reports the de-quantization time complexity and extra bits per parameter (bpm) needed for a linear layer $f(\mathbf{X}) = \mathbf{X}\mathbf{W}$, where $\mathbf{X}$ has shape $n \times d$ and $\mathbf{W}$ has shape $d \times c$. We compare the overhead of RabitQ and RabitQ-H against the cost of matrix multiplication in the original (non-quantized) model. The analysis reveals that, with RabitQ-H, both the extra time and memory overhead introduced by quantization are negligible since they are on the order of $O(1/c)$ or $O(\log d/c)$ relative to the original model. In contrast, the original RabitQ introduces an overhead of $\Theta(d/c)$, which becomes prohibitively large in practical settings where $d \approx c$.

## 7 DISCUSSION AND CONCLUSION

In this work, we introduce RaanA: a new PTQ framework combining RaBitQ-H, a variant of RaBitQ that especially fits LLM quantization, and AllocateBits, an algorithm to allocate bit-widths across layers optimally. RaanA overcomes traditional challenges of PTQ methods such as high reliance on calibration and inflexible bits allocation. Extensive experiment results validate the performance of RaanA, especially highlighting the effectiveness of zero-shot calibration, eliminating the requirement of heavy calibration.

**Limitations and Future Work** Here we discuss current limitations of the RaanA framework and potential future directions to improve them. 1) More efficient implementation: As we mentioned in Section 6.3, currently the implementation of RaanA is not optimal, as the computation is bottlenecked by the CPU-bound execution of RaBitQ. A more efficient implementation / approximation of RaBitQ (ideally on GPU) would vastly accelerates RaanA. 2) Finer-grained bits-allocation: currently RaanA allocates bit-widths layer-wisely, constraining the parameters in each layer to share the same bit-width, which can be sub-optimal. It is possible to consider a finer-grained bits-allocation, e.g. column-wisely or even entry-wisely, as a future direction.

**Remark on LLM Design** Last but not least, we would like to advocate future large language model designers to use a powers of 2 as the hidden size more often; as arbitrary as it may seem, this choice actually has the following advantages that make quantization easier and faster: 1) It maximizes the GCD between layers and thus improves the speed of the AllocateBits algorithm. 2) It makes it easier to use fast Hadamard Transformations since fast Hadamard Transformation is only defined for spaces whose dimension is a power of 2.

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

# A  DETAILED INTRODUCTION TO ALGORITHMIC TOOLS

In this section, we introduce the algorithms used in RaanA.

## A.1  DETAILED INTRODUCTION TO RANDOMIZED HADAMARD TRANSFORMATION

In this sub-section, we provide a formal definition of RHT. Let $d$ be a positive integer number $d$ that is a power of 2, the Hadamard Transformation of size $d$ is a linear transformation recursively defined by the following matrix:

$$\boldsymbol{H}_d := \begin{bmatrix} \boldsymbol{H}_{d/2} & \boldsymbol{H}_{d/2} \\ \boldsymbol{H}_{d/2} & -\boldsymbol{H}_{d/2} \end{bmatrix} \tag{6}$$

and $\boldsymbol{H}_1 = 1$. For a vector $\boldsymbol{x} \in \mathbb{R}^d$, we define

$$\mathsf{Hadamard}(\boldsymbol{x}) := \frac{1}{\sqrt{d}} \boldsymbol{H}_d \boldsymbol{x}. \tag{7}$$

it has been shown that the Hadamard Transformation can be computed in a fast way, i.e. applying a Hadamard Transformation to each column of an $d \times n$ matrix only requires $O(n \log d)$ time.

Let $\boldsymbol{\xi} = \{\xi_i\}_{i=1}^n$ be $n$ i.i.d. sampled Rademacher variable, and let $\boldsymbol{D} = \mathrm{diag}\,(\boldsymbol{\xi})$. For a matrix $\boldsymbol{x} \in \mathbb{R}^d$, we say

$$\boldsymbol{x} \mapsto \mathsf{Hadamard}\,(\boldsymbol{D}\boldsymbol{x}) \tag{8}$$

the Randomized Hadamard Transformation (RHT) of $\boldsymbol{x}$, which is exactly what we used in Algorithm 2. Since the Hadamard Transformation is orthonormal, it's not hard to restore the result of RHT, we only need to store the vector $\boldsymbol{\zeta}$, which has only $d$ binary entries.

## A.2  DETAILED INTRODUCTION TO RABITQ

In this sub-section, we provide a brief introduction to RaBitQ (Gao & Long, 2024) and its multi-bit extension (Gao et al., 2024). The RaBitQ methods were proposed initially for vector quantization in database systems. They target to produce accurate estimation of inner product and Euclidean distances based on the quantized vectors while using the minimum space for storing quantization codes. Specifically, let $\boldsymbol{P}$ be a Johnson-Lindenstrauss Transformation (a.k.a., random rotation) (Johnson & Lindenstrauss, 1984). For a vector $\boldsymbol{x} \in \mathbb{R}^d$, RaBitQ randomly rotates it into $\boldsymbol{P}\boldsymbol{x}$ and quantizes $\boldsymbol{P}\boldsymbol{x}$ to a vector of $b$-bit unsigned integers $\bar{\boldsymbol{x}} \in [2^b]^d$ with a rescaling factor $t \in \mathbb{R}$. Then for another vector $\boldsymbol{y} \in \mathbb{R}^d$, RaBitQ estimates the inner product $\langle \boldsymbol{x}, \boldsymbol{y} \rangle$ with

$$\langle \boldsymbol{x}, \boldsymbol{y} \rangle \approx \langle t \cdot (\bar{\boldsymbol{x}} - c_b \cdot \boldsymbol{1}_d), \boldsymbol{P}\boldsymbol{y} \rangle \tag{9}$$

where $\boldsymbol{1}_d$ denotes the $d$-dimensional vector whose coordinates are ones and $c_b = (2^b - 1)/2$. For the details of the quantization algorithms and the rescaling factors, we refer readers to the original paper (Gao et al., 2024).

As has been proven in the original RaBitQ papers, RaBitQ guarantees that the estimation is **unbiased** and **asymptotically optimal** in terms of the trade-off between the error bound and the space for storing the codes. Specifically, with probability as least $1 - \delta$, to guarantee that

$$\big| \langle \boldsymbol{x}, \boldsymbol{y} \rangle - \langle t(\bar{\boldsymbol{x}} - c_b \boldsymbol{1}_d), \boldsymbol{P}\boldsymbol{y} \rangle \big| < \epsilon \|\boldsymbol{x}\| \|\boldsymbol{y}\| \tag{10}$$

it suffices to let $b = \Theta\left(\log\left(\frac{1}{d} \cdot \frac{\log(1/\delta)}{\epsilon^2}\right)\right)$ when $\epsilon$ is sufficiently small, i.e., $\frac{\log(1/\delta)}{\epsilon^2} > d$. This result achieves the optimality established in a theoretical study (Alon & Klartag, 2017).

Additionally, RaBitQ provides an empirical formula for the trade-off between errors and spaces. Specifically, with probability at least 99.9%, we have

$$\big| \langle \boldsymbol{x}, \boldsymbol{y} \rangle - \langle t(\bar{\boldsymbol{x}} - c_b \boldsymbol{1}_d), \boldsymbol{P}\boldsymbol{y} \rangle \big| < \frac{c_{\mathrm{error}}}{\sqrt{d}2^b} \|\boldsymbol{x}\| \|\boldsymbol{y}\| \tag{11}$$

where $c_{\mathrm{error}} = 5.75$.

## B  THEORETICAL RESULTS

In this section, we prove the main theorem of this paper, which directly leads to Theorem 4.2. We first state a formal version of Theorem 4.1, stating that the error from RaBitQ (and RaBitQ-H) quantization has sub-exponential error.

**Assumption B.1.** [(Gao et al., 2024)] There exists a constant $K > 0$, such that for any $k \in [L], i \in [n], j \in [c_k]$, $\boldsymbol{\epsilon}^{(k)}(b) \in \mathbb{R}^{d_k}$ is a random vector such that

$$\forall t > 0, \mathbb{P}\left\{\left|\epsilon_{i,j}^{(k)}(b)\right| > t\right\} \leq 2\exp\left[-K\left(\frac{t\sqrt{d_k}}{2^{-b_k}\left\|\boldsymbol{x}_i^{(k)}\right\|\left\|\boldsymbol{w}_j^{(k)}\right\|}\right)^2\right]. \tag{12}$$

Notice that Theorem B.1 does not assume $\epsilon_{i,j}^{(k)}(b)$-s are independent. We provide the following algorithm analyzing the behavior of a function under an $O(1/d)$ small error, which together with Theorem B.1 implies Theorem 4.2.

**Theorem B.2.** *There exists constant $K > 0$ satisfies the following statement. Suppose $c \geq 2$, $d \gg c$ and $\boldsymbol{\epsilon} \in \mathbb{R}^{n \times c}$ is a random vector, such that $\forall i, j \in [n] \times [c]$,*

$$\mathbb{P}\{|\epsilon_{i,j}| > t\} \leq 2\exp\left(-\frac{Ct^2 d}{\lambda^2 \gamma_{i,j}^2}\right) \tag{13}$$

*for some constant $K_1$ (notice that $\epsilon_{i,j}$-s are not necessarily independent). Let $g : \mathbb{R}^{n \times c} \to \mathbb{R}$ be a smooth function, then for any $\boldsymbol{h} \in \mathbb{R}^{n \times c}$, the following statement holds with probability at least $0.99$:*

$$|g(\boldsymbol{h}) - g(\boldsymbol{h} + \boldsymbol{\epsilon})| \leq K\sqrt{\frac{\log c}{d}}\|\boldsymbol{\gamma}\|_{\mathcal{F}}\|\nabla g(\boldsymbol{h})\|_{\mathcal{F}} + O\left(\frac{\|\nabla^2 g(\boldsymbol{h})\|}{d}\right). \tag{14}$$

*Proof.* From the given condition, we have

$$\forall t > 0, \mathbb{P}\{|\epsilon_{i,j}| > t\gamma_{i,j}\} \leq 2\exp\left(-K_1 t^2 d/\lambda^2\right). \tag{15}$$

Thus we have

$$\forall t > 0, \mathbb{P}\{\exists i, j \in [n] \times [c], |\epsilon_{i,j}| > t\gamma_{i,j}\} \leq 2c\exp\left(-K_1 t^2 d/\lambda^2\right). \tag{16}$$

For constant $K_2 > 0$, let $t_0 = K_2\lambda\sqrt{\frac{\log c}{d}}$. It is evident that there exists a constant $K_2$ that only depends on $K_1$, such that

$$\mathbb{P}\{\exists i, j \in [n] \times [c], |\epsilon_{i,j}| > t_0\gamma_{i,j}\} \leq 0.01. \tag{17}$$

Thus, with probability $\geq 0.99$, we have

$$\forall i, j \in [n] \times [c], |\epsilon_{i,j}| \leq K_2\lambda\gamma_{i,j}\sqrt{\frac{\log c}{d}}. \tag{18}$$

Therefore, with probability at least $0.99$, we have

$$\|\boldsymbol{\epsilon}\|_{\mathcal{F}} \leq \sqrt{\sum_{i=1}^{n}\sum_{j=1}^{c}\epsilon_{i,j}^2} \leq K_2\lambda\sqrt{\frac{\log c}{d}}\|\boldsymbol{\gamma}\|_{\mathcal{F}}. \tag{19}$$

Using Taylor expansion, we have

$$|g(\boldsymbol{h}) - g(\boldsymbol{h} + \boldsymbol{\epsilon})| \leq \langle \nabla g(\boldsymbol{h}), \boldsymbol{\epsilon} \rangle + O\left(\|\nabla^2 g(\boldsymbol{h})\|\|\boldsymbol{\epsilon}\|^2\right) \tag{20}$$

$$\leq \|\nabla g(\boldsymbol{h})\|_{\mathcal{F}}\|\boldsymbol{\epsilon}\|_{\mathcal{F}} + O\left(\frac{\|\nabla^2 g(\boldsymbol{h})\|}{d}\right) \tag{21}$$

$$\leq K_2\lambda\sqrt{\frac{\log c}{d}}\|\boldsymbol{\gamma}\|_{\mathcal{F}}\|\nabla g(\boldsymbol{h})\|_{\mathcal{F}} + O\left(\frac{\|\nabla^2 g(\boldsymbol{h})\|}{d}\right). \tag{22}$$

$\square$

Theorem 4.2 is thus a direct corollary of Theorem B.1 and Theorem B.2. Notice that in Theorem 4.2 we view the second-order derivative of $f$ w.r.t. $\boldsymbol{H}_k$ a constant (evaluated at a fixed point), and therefore omit the $\|\nabla^2 \cdot \|$ term.

## C    IMPLEMENTATION DETAILS

In this section, we provide some implementation details that are not elaborated in the main text due to space limit.

### C.1    BITS-ALLOCATION ALGORITHM

In Section 4, we mentioned that the bits-allocation problem can be solved efficiently by a dynamic programming algorithm after applying the divide-by-GCD trick. In Algorithm 4 we provide the detailed algorithm description.

In our implementation of Algorithm 4, we compute $\alpha_k$ as

$$\alpha_k = \frac{1}{\sqrt{d_k}} \left\| \frac{\partial f(\boldsymbol{\theta}, \boldsymbol{x})}{\partial \boldsymbol{H}^{(k)}} \Big|_{\substack{\boldsymbol{x}=\boldsymbol{x} \\ \boldsymbol{\theta}=\boldsymbol{\theta}}} \right\|_{\mathcal{F}} \left\| \boldsymbol{X}^{(k)} \right\|_{\mathcal{F}} \left\| \boldsymbol{W}^{(k)} \right\|_{\mathcal{F}}, \tag{23}$$

omitting the $\log c_k$ term in the , since it is almost constant across layers and therefore has negligible impact on the optimization.

---

**Algorithm 4:** Bits Allocation

---

**Input:** Coefficients $\{\alpha_k\}_{k=1}^L \in \mathbb{R}^L$, number of bits candidate $\mathscr{B}$, overall budget $R \in \mathbb{N}$
Initialize $f_{k,r} = +\infty$, where $k \in [L], r \in [R] \cup \{0\}$;
$g \leftarrow \gcd(m_1, \cdots, m_L, R)$;
**for** $k = 1, 2, \cdots L$ **do**
    **for** $b \in \mathscr{B}$ **do**
        $r \leftarrow \left\lfloor \frac{m_k B}{g} + \frac{1}{2} \right\rfloor$;
        $c \leftarrow \alpha_k 2^{-b}$;
        **if** $k = 1$ **then**
            $f_{k,r} \leftarrow c$;
            $s_{k,r} \leftarrow \{b\}$;
        **end**
        **else**
            **for** $r' = 0, 1, \cdots \frac{R}{g} - r$ **do**
                **if** $f_{k,r'+r} > f_{k-1,r'} + c_k$ **then**
                    $f_{k,r'+r} \leftarrow f_{k-1,r'} + c_k$;
                    $s_{k,r'+r} \leftarrow s_{k,r'} \parallel \{b\}$
                **end**
            **end**
        **end**
    **end**
**end**
$r^* \leftarrow \arg\min_{r=0}^{R} f_{L,r}$ ;
**Return:** $s_{L,r^*}$.

---

**Algorithm 4: The algorithm for bits allocation**. In the algorithm $\alpha_k = \mathbb{E}_{x \sim \mathcal{D}} \lambda_k r_k$ is the coefficient for the $k$-th layer, $\{b\}$ represents a sequence with only one element $b$ and $\parallel$ stands for sequence concatenation. The returned value is a sequence indicating the optimal bit-widths each layer, i.e. $s_{L,r^*} = \{b_k^*\}_{k=1}^L$.

### C.2    RANDOMIZED HADAMARD TRANSFORMATION FOR ARBITRARY DIMENSIONALITY

As we mentioned in Section A.1, the fast Hadamard Transformation is only defined with vector dimensionality $d$ that is a power of 2. However, in practice, it is not always satisfied. In previous work such as (Tseng et al., 2024), this issue is solved by finding the largest factor of $d$ which is a power of 2, say $\tilde{d}$, and applying Hadamard Transformation block-wisely, with each block has size $\tilde{d}$.

However, in our experiment, we found this method extremely inefficient. For example, for LLaMA models, there can be $> 20$ blocks.

Therefore, in this paper, we apply an easy and universal method to address this issue. We first find the largest power of 2 that is less or equal to $d$, i.e. $\hat{d} = 2^{\lfloor \log_2 d \rfloor}$, and apply RHT for the first and last $\hat{d}$ dimensions respectively. Our algorithm is described in Algorithm 5.

---

**Algorithm 5:** Practical RHT

**Input:** Vector $\boldsymbol{x} \in \mathbb{R}^d$

$\hat{d} = 2^{\lfloor \log_2 d \rfloor}$;

**for** $j = 1,2$ **do**

$\quad \boldsymbol{\xi}^{(j)} \leftarrow \left\{ \xi_i^{(j)} \right\}_{i=1}^{\hat{d}}$, where $\xi_i^{(j)}$ is sampled i.i.d. from the Rademacher distribution;

$\quad \boldsymbol{D}^{(j)} \leftarrow \mathrm{diag}\left( \boldsymbol{\xi}^{(j)} \right)$;

**end**

$\boldsymbol{x}_{1:\hat{d}} \leftarrow \mathsf{Hadamard}\left( \boldsymbol{D}^{(1)} \boldsymbol{x}_{1:d} \right)$;

$\boldsymbol{x}_{d-\hat{d}+1:d} \leftarrow \mathsf{Hadamard}\left( \boldsymbol{D}^{(2)} \boldsymbol{x}_{d-\hat{d}+1:d} \right)$;

**Return:** $\left( \boldsymbol{x}, \boldsymbol{D}^{(1)}, \boldsymbol{D}^{(2)} \right)$.

---

**Algorithm 4: Practical Randomized Hadamard Transformation**. $\boldsymbol{x}_{a:b}$ refers to the sub-vector of $\boldsymbol{x}$ consists of the $a$-th entry to the $b$-th entry of $\boldsymbol{x}$.

### C.3 TRICKS USED IN QUANTIZATION

In the actual implementation of RaanA, we optionally apply some transformations before performing quantization. Formally, for the $d \times c$ matrix, we define a **trick** to be a invertible **linear** transformation $T : \mathbb{R}^{n \times d} \to \mathbb{R}^{n \times d}$. Then for a linear layer where input matrix is $\boldsymbol{X} \in \mathbb{R}^{n \times d}$ and weight matrix is $\boldsymbol{W} \in \mathbb{R}^{d \times c}$, we have

$$\boldsymbol{X}\boldsymbol{W} = T^{-1}\left( T(\boldsymbol{X})\boldsymbol{W} \right). \tag{24}$$

Notice that $T$ can be have a memory, i.e. it can return an auxiliary term to help $T^{-1}$ to recover the computation result. After applying $T$, in the de-quantization stage we only need to estimate the matrix multiplication results for $T(\boldsymbol{X})\boldsymbol{W}$.

In practice, we the following heuristic tricks are optionally used.

- **Centralization**: $T(\boldsymbol{X}) = \left[ \boldsymbol{X} - \boldsymbol{1}s(\boldsymbol{X})^\top, s\left( \boldsymbol{X} \right) \right]$, where $s(\boldsymbol{X}) \in \mathbb{R}^d$ is the average of all rows of $\boldsymbol{X}$;

- **Row Outlier Excluding**: $T(\boldsymbol{X}) = [\boldsymbol{X}_{\neg \boldsymbol{M}_r}, \boldsymbol{X}_{\boldsymbol{M}_r}]$, where $\boldsymbol{M}_r$ is a mask vector selecting the top $0.3\%$ rows of $\boldsymbol{X}$ with largest norm, and $\neg \boldsymbol{M}_r$ selects the opposite. $\boldsymbol{X}_{\boldsymbol{M}_r}$ indicates selecting rows of $\boldsymbol{X}$ according to the mask vector $\boldsymbol{M}_r$;

- **Column Outlier Excluding**: $T(\boldsymbol{X}) = [\boldsymbol{X}_{:,\boldsymbol{M}_c}, \boldsymbol{X}_{:,\neg \boldsymbol{M}_c}]$, where $\boldsymbol{M}_c$ is a mask vector selecting the top $0.3\%$ columns of $\boldsymbol{X}$ with largest norm, and $\neg \boldsymbol{M}_c$ selects the opposite. $\boldsymbol{X}_{:,\boldsymbol{M}_c}$ indicates selecting columns of $\boldsymbol{X}$ according to the mask vector $\boldsymbol{M}_c$.

For each of the trick functions $T$ described above, it returns two values. The first return value is the matrix that joins the subsequent computation, and the second return value is to be memorized in order to recover the original computational result.

Notice that for Row Outlier Excluding and Column Outlier Excluding, the trick needs to store a few rows / columns of $\boldsymbol{X}$. We intentionally restrict the outlier ratios less than $0.3\%$ in order to keep the extra bits used to store the extra information negligible.

Practically we find these tricks have incremental contribution on the performance, and different combinations of them perform differently across settings. Below in Section C.4 we perform a preliminary experiment on their impact to the performance. In all the other experiments presented in this paper, to keep the configuration consistent and avoid heavy hyper-parameter tuning, we use Centralization and Column Outlier Excluding.

## C.4 ABLATION STUDIES ON IMPLEMENTATION TRICKS

In order to show the contribution of the implementation tricks to the performance, in this subsection we compare their impact to the performance with different combinations. The results are shown in Table 6. It can be easily inferred from the table that these implementation tricks only have incremental contribution to the overall performance.

| Combination of tricks | few (3.3bits) | zero (3.3bits) | few (4.3bits) | zero (4.3bits) |
|---|---|---|---|---|
| (none) | 11.35 | 12.20 | 10.35 | 10.49 |
| cent | 11.14 | 11.93 | 10.33 | 10.45 |
| cent, col-out | 11.05 | 11.56 | **10.06** | 10.27 |
| cent, row-out | **10.98** | 11.83 | 10.24 | 10.51 |
| cent, col-out, row-out | 11.02 | **11.49** | 10.14 | **10.25** |
| col-out | 11.18 | 11.71 | 10.21 | 10.32 |
| row-out | 11.24 | 11.93 | 10.23 | 10.37 |
| col-out, row-out | 11.13 | 11.63 | 10.24 | 10.32 |

Table 6: **Perplexity comparison between different combinations of the implementation tricks**. All experiments are performed with qwen-8b on wikitext2. In the table, "cent" refers to Centralization, "row-out" refers to Row Outlier Excluding and "col-out" refers to Column Outlier Excluding.

## D ADDITIONAL EXPERIMENT RESULTS

In this section we present additional experiment results on extra models / datasets. These additional experimental results clearly support our claim in main paper.

Table 7 presents the perplexity results of RaanA with Qwen3 on wikitext2 dataset, compared with GPTQ and AWQ with group size 128. The baseline results are taken from (Zheng et al., 2025).

| Method | Avg. bits | qwen-4b | qwen-8b | qwen-14b |
|---|---|---|---|---|
| fp16 | 16 | 13.7 | 9.71 | 8.64 |
| AWQ | 2+ | 1.38E7 | 1.21E7 | 6.06E6 |
| GPTQ | 2+ | **1.95E2** | 52.1 | 25.5 |
| RaanA | 2.3 | 328.20 | **38.13** | **17.48** |
| AWQ | 3+ | 91.0 | 27.5 | 19.4 |
| RaanA | 3.1 | 20.40 | 12.05 | 10.00 |
| RaanA | 3.3 | **18.63** | **11.14** | **9.79** |
| AWQ | 4+ | 18.1 | 11.3 | 9.48 |
| GPTQ | 4+ | **13.5** | **9.96** | **8.88** |
| RaanA | 4.1 | 15.4 | 10.18 | 9.23 |
| RaanA | 4.3 | 14.9 | 9.97 | 9.18 |

Table 7: **Perplexity results of Qwen3 on wikitext2**. The setting and format are the same as Table 1, except model.

| Method | Avg. bits | llama-7b | llama-13b | llama-7b | llama-13b |
|--------|-----------|----------|-----------|----------|-----------|
| fp16 | 16 | 7.08 | 6.61 | 6.97 | 6.47 |
| GPTQ | 2+ | 27.71 | 15.29 | 33.70 | NAN |
| AWQ | 2+ | 11.9e5 | 2.3e5 | 1.7e5 | 9.4e4 |
| Quip#* | 2 | **11.7** | **8.67** | 14.8 | **9.57** |
| OmniQuant | 2+ | 12.97 | 10.36 | 15.02 | 11.05 |
| RaanA | 2.1 | 20.88 | 13.08 | 23.17 | - |
| RaanA | 2.3 | 12.96 | 9.18 | **12.66** | 10.37 |
| GPTQ | 3+ | 7.85 | 7.10 | 7.89 | 7.00 |
| AWQ | 3+ | 7.92 | 7.07 | 7.84 | **6.94** |
| OmniQuant | 3+ | **7.75** | 7.05 | **7.75** | 6.98 |
| Quip#* | 3 | 7.82 | **6.98** | 7.85 | 6.98 |
| RaanA | 3.1 | 8.25 | 7.34 | 8.38 | 7.52 |
| RaanA | 3.3 | 7.92 | 7.15 | 7.99 | - |
| EasyQuant | 4+ | 7.71 | 6.97 | - | - |
| OmniQuant | 4+ | **7.21** | **6.69** | 7.12 | **6.56** |
| GPTQ | 4+ | 7.21 | 6.69 | 7.12 | 6.56 |
| AWQ | 4+ | 7.21 | 6.70 | 7.13 | 6.56 |
| Quip#* | 4 | 7.25 | 6.70 | 7.17 | 6.59 |
| RaanA | 4.1 | 7.51 | 6.91 | 7.53 | 6.88 |
| RaanA | 4.3 | 7.52 | 6.87 | 7.45 | 6.83 |

Table 8: **Perplexity results on c4**. The setting and format are the same as Table 1, except dataset.

Table 8 contains perplexity comparison between RaanA with baseline methods with LLaMA models on c4, where RaanA uses few-shot calibration. Table 9 contains the few-shot vs zero-shot comparison of RaanA on c4.

| Method | Avg. bits | llama-7b | llama-13b | llama-7b | llama-13b |
|--------|-----------|----------|-----------|----------|-----------|
| fp16 | 16 | 7.08 | 6.61 | 6.97 | 6.47 |
| RaanA-zero | 2.1 | 19.37 | 13.14 | 31.31 | 18.03 |
| RaanA-few | 2.1 | 20.88 | 13.08 | 23.17 | - |
| RaanA-zero | 3.1 | 8.44 | 7.55 | 8.61 | 7.69 |
| RaanA-few | 3.1 | 8.25 | 7.34 | 8.38 | 7.52 |
| RaanA-zero | 4.1 | 7.56 | 6.96 | 7.59 | 6.93 |
| RaanA-few | 4.1 | 7.51 | 6.91 | 7.53 | 6.88 |

Table 9: **Perplexity Comparison Between Zero-shot Calibration and Few-shot Calibration on c4**. The setting and format are the same as Table 2, except dataset.

## LARGE LANGUAGE MODEL USAGE

In preparing this submission, we used a large language model (ChatGPT) as an assistive tool for language polishing. Moreover, the synthetic sentence used in the zero-shot calibration setting was suggested by ChatGPT (mentioned in Section 4.2). Apart from this sentence and aforementioned language polishing, the model did not contribute to the research ideation, experimental design, data analysis, or the generation of scientific content. All substantive content, results, and conclusions presented in this paper were conceived, written, and verified by the authors, who take full responsibility for the work.

