# OpenReview forum: "RaanA: A Fast, Flexible, and Data-Efficient Post-Training Quantization Algorithm"
_ICLR.cc/2026/Conference — ICLR 2026 Conference Withdrawn Submission_

### Official Review · Reviewer_vcE7 · 2025-10-30

**Soundness:** 3
**Presentation:** 3
**Contribution:** 2
**Rating:** 4
**Confidence:** 4

**Summary:**

This paper introduces RaanA, a novel post-training quantization (PTQ) framework designed to enhance the inference efficiency of large language models (LLMs).
RaanA addresses key limitations of existing PTQ methods, such as heavy reliance on calibration data and inflexible bit-width allocation by integrating two core components:
(1) RaBitQ-H: A highly efficient variant of the RaBitQ vector quantization algorithm, adapted for LLMs by replacing the computationally expensive random rotation with a Randomized Hadamard Transformation (RHT).
(2) AllocateBits: An optimal bit allocation algorithm that formulates bit-width assignment across layers as an integer programming problem, dynamically solved via dynamic programming.

**Strengths:**

This paper demonstrates significant originality by creatively bridging advanced vector quantization techniques from database systems (RaBitQ) with the practical demands of LLM PTQ.
The development of RaBitQ-H is a key innovation, replacing a computationally prohibitive random rotation with a Randomized Hadamard Transformation to make the method viable for high-dimensional model weights. The AllocateBits algorithm also presents a novel, principled approach to mixed-precision quantization, moving beyond simple heuristics to an optimal, solvable integer program.

The quality of the work is high, supported by extensive and rigorous experiments on major model families (LLaMA, Qwen) across multiple bit-widths. The results are highly competitive with state-of-the-art methods, particularly in the challenging ultra-low-bit regime (~2 bits).

The clarity is commendable; the paper is well-structured, and the algorithmic contributions are precisely defined, with a clear separation of the overall framework, bit allocation, and core quantization components.

Its significance is substantial. RaanA directly tackles critical deployment barriers for LLMs—high computational cost and data dependency—by offering a fast, data-efficient, and flexible solution. The demonstration of effective zero-shot calibration is especially impactful, potentially removing the need for carefully curated calibration datasets altogether and enhancing the practicality and accessibility of LLM quantization.

**Weaknesses:**

The paper's primary weakness is the suboptimal implementation of its core algorithm.

(1) RaBitQ is a CPU-bound bottleneck contradicts the goal of a "fast" framework and limits practical utility. A GPU implementation is crucial for true speed competitiveness.

(2) The evaluation could be more comprehensive. While perplexity is standard, the absence of inference latency and memory footprint measurements on actual hardware is a significant omission for a method claiming efficiency gains. Reporting wall-clock quantization time for baselines would also contextualize the claimed speed advantage.

(3) The layer-wise bit allocation, while an improvement, is noted as a sub-optimal constraint itself. The authors identify finer-grained (e.g., column-wise) allocation as future work, but not exploring a simple, coarser alternative (e.g., grouping layers by type/sensitivity) leaves a clear, actionable avenue for immediate improvement unexplored in the current evaluation.

**Questions:**

There are two questions:

(1) What are the fundamental algorithmic operations in RaBitQ that make it challenging to port to GPU, and are there known parallelization strategies (e.g., using CUDA kernels for the core quantization steps) that could be applied?

(2) Could you provide measurements of the end-to-end inference latency (e.g., time to generate the first token and time per subsequent token) for RaanA-quantized models compared to the fp16 baseline and a key baseline like GPTQ or Quip#? This should be done on a standard hardware setup (e.g., a single A100).

---

### Official Review · Reviewer_xLVw · 2025-10-31

**Soundness:** 3
**Presentation:** 3
**Contribution:** 2
**Rating:** 2
**Confidence:** 4

**Summary:**

This paper proposes RaanA, a novel post-training quantization (PTQ) framework for large language models. The framework combines two main components: (1) RaBitQ-H, a variant of the RaBitQ vector quantization algorithm adapted for LLM quantization by replacing expensive random rotations with efficient Randomized Hadamard Transformations, and (2) AllocateBits, an algorithm that optimally allocates bit-widths across layers based on estimated quantization sensitivity using dynamic programming. The authors evaluate RaanA on LLaMA and Qwen models using perplexity on WikiText2 and C4 datasets. Results show RaanA achieves comparable or better perplexity, particularly in the extreme 2-bit regime, while being significantly faster and requiring far less calibration data.

**Strengths:**

- This paper proposes RaaanA, which can quickly determine the bit allocation for each layer using only a small amount of calibration data. Compared with heuristic methods, it also offers better interpretability.
- RaanA requires very few quantization resources. For example, it takes only 301.74 seconds to complete the entire PTQ process for LLaMA2-7B.

**Weaknesses:**

- This paper lacks novelty, as the proposed Randomized Hadamard Transformation (RHT) has already been widely used in quantization[1] and is considered a common trick.
- This paper only reports perplexity experiments on Wikitext2 and a C4 subset, lacking broader evaluations such as MMLU and math. As a result, it fails to demonstrate that the proposed method does not suffer from overfitting when using a small calibration dataset.
- As a weight-only mixed-precision quantization method, the experimental results presented in this paper do not outperform previous approaches. For instance, in Table 1, RaanA fails to surpass OmniQuant at average bit levels of 2.1, 3.1, and 4.1 (note that OmniQuant uses a group size of 128, with average bits of 2.125, 3.125, and 4.125). Moreover, OmniQuant was published two years ago, and this paper does not include comparisons with more recent methods[2].

[1] Ashkboos, Saleh, et al. "Quarot: Outlier-free 4-bit inference in rotated llms." Advances in Neural Information Processing Systems 37 (2024): 100213-100240.

[2] Li, Yuhang, et al. "GPTAQ: Efficient Finetuning-Free Quantization for Asymmetric Calibration."

**Questions:**

- Have you compared the results of RaanA and other methods, such as OmniQuant, using the same calibration dataset?
- For other issues, please refer to the Weaknesses section.

---

### Official Review · Reviewer_78zZ · 2025-10-31

**Soundness:** 1
**Presentation:** 1
**Contribution:** 1
**Rating:** 2
**Confidence:** 4

**Summary:**

The paper tackles post training quantization by combining RabitQ and AllocateBits. In doing so they formulate a vector quantization approach that utilizes per layer mixed precision.
strengths:

**Strengths:**

- the empirical results seem promising.

**Weaknesses:**

- the notation of Assumption 4.1 (which is not really an assumption) is inappropriate. Several symbols are used without being defined. The statement is based on some K>0 that does not appear in the theorem/assumption.
- Corollary 4.2 is unprofessionally stated. High probability events should not be denoted as having a probability of "at least 0.99" which is a subjective amount and can be either good or bad depending on the context. A proper way to introduce high probability events is by lower bounding the probability by 1-\delta, where \delta is a scalr that actually affects the bound on the error.
- the gcd is not needed for (5) to hold. Is this decorative math?
- overall the paper is poorly written and its clarity should be significantly improved.

**Questions:**

- Can the mixed precision be done intra-layer such as in FGMP (Hooper et al.)?

---

### Note · Authors · 2025-11-24

**Comment:**

We have decided to withdraw our submission. We sincerely thank the reviewers and the program committee for their time and effort.

**Withdrawal Confirmation:**

I have read and agree with the venue's withdrawal policy on behalf of myself and my co-authors.